# ParaDiS: Parallelly Distributable Slimmable Neural Networks

## Abstract

When several limited power devices are available, one of the most efficient ways to make profit of these resources, while reducing the processing latency and communication load, is to run in parallel several neural sub-networks and to fuse the result at the end of processing. However, such a combination of sub-networks must be trained specifically for each particular configuration of devices (characterized by number of devices and their capacities) which may vary over different model deployments and even within the same deployment. In this work we introduce *parallelly distributable slimmable (ParaDiS) neural networks* that are splittable in parallel among various device configurations without retraining. While inspired by slimmable networks allowing instant adaptation to resources on just one device, ParaDiS networks consist of several multi-device distributable configurations or switches that strongly share the parameters between them. We evaluate ParaDiS framework on MobileNet v1 and ResNet-50 architectures on ImageNet classification task and WDSR architecture for image super-resolution task. We show that ParaDiS switches achieve similar or better accuracy than the individual models, i.e., distributed models of the same structure trained individually. Moreover, we show that, as compared to universally slimmable networks that are not distributable, the accuracy of distributable ParaDiS switches either does not drop at all or drops by a maximum of 1 % only in the worst cases. Finally, once distributed over several devices, ParaDiS outperforms greatly slimmable models.

## 1 Introduction

Neural networks are more and more frequently run on end user devices such as mobile phones or Internet of Things (IoT) devices instead of the cloud. This is due to clear advantages in terms of better processing latency and privacy preservation. However, those devices have often very limited (runtime and memory) resources. While manually designing lightweight networks (Howard et al., 2017; Zhang et al., 2018) or using model compression schemes (Cheng et al., 2018) (e.g., pruning (LeCun et al., 1990) or knowledge distillation (Hinton et al., 2015; Xie et al., 2020)) allow satisfying particular constraints, these approaches are not well suited when those resources vary across devices and over time. Indeed, a particular model is handcrafted or compressed for given resource constraints, and once those constraints change, a new model needs to be created. To this end, a new family of approaches that are able to instantly trade off between the accuracy and efficiency was introduced. We call these approaches (or models) *flexible* (Ozerov & Duong, 2021), and they include among others early-exit models (Huang et al., 2018), slimmable neural networks (Yu et al., 2019; Yu & Huang, 2019), Once-for-All (OFA) networks (Cai et al., 2019; Sahni et al., 2021), and neural mixture models (Ruiz & Verbeek, 2020).

An alternative path to gain in efficiency under constrained resources setting, consists in distributing the neural network inference over several devices (if available) (Teerapittayanon et al., 2017). The simplest way is to distribute a neural network sequentially, e.g., in case of two devices by executing early layers on the first device and the remaining layers on the second one, while transmitting the intermediate features between the two devices over a network (see Fig. 1 (B)). Depending on the transmission network bandwidth, the features might undergo a lossless or a more or less severe lossy compression (Choi et al., 2018). Neural network (A) (see Fig. 1) might be either distributed as it is or, in case the features are distorted by compression, after a fine-tuning or retraining (Choi et al., 2018). Though sequential distribution might overcome issues of limited memory and com-

puting power, it does not decrease the processing latency and might only increase it because of the additional communication delay. This might be overcome via a parallel distribution (Hadidi et al., 2019), as illustrated on Fig. 1 (C). However, to maintain the functionality of the original neural network all convolutional and fully connected layers need to communicate, which requires a considerable communication burden and increases processing latency. To overcome this the model may be distributed parallelly without intermediate communication (Bhardwaj et al., 2019; Asif et al., 2019), as on Fig. 1 (D), where the only communication needed is to transmit the data to each device and the final result from each device for a fusion. In this work we chose to focus on this parallel distribution without communication and call it simply *parallel distribution* hereafter, since it leads to a much smaller processing latency, as compared to sequential distribution; does not suffer from high communication burden of communicating parallel distribution, and is also easy to deploy. The price to be paid for these advantages consists in a potentially decreasing performance and a need of retraining due to the loss of connections between the intermediate layers.

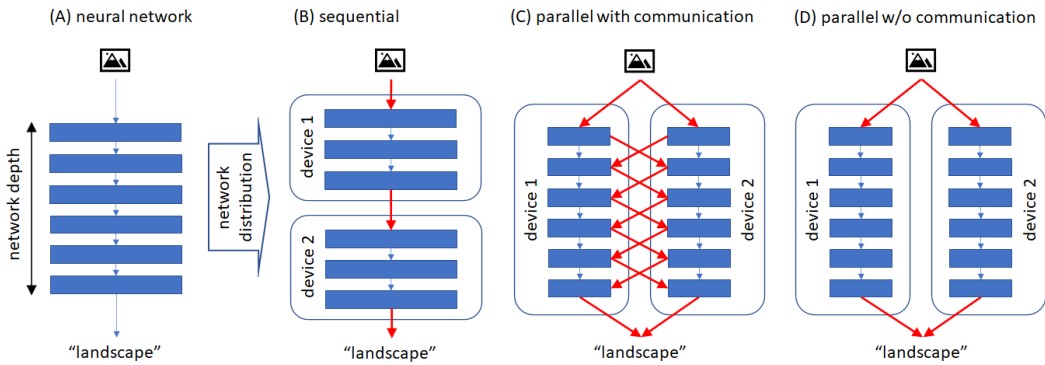

Figure 1: Different ways of distributing a neural network (A) over two devices: sequential distribution (B), parallel distribution with communication (C), and parallel distribution without communication (D). Bold red arrows indicate communication over the transmission network.

The question we are trying to answer in this work is: *Can we have one neural network that may instantly and near-optimally [1] be parallelly distributed over several devices, regardless of the number of devices and their capacities?* In this paper we introduce a new framework called *parallelly distributable slimmable neural networks* or *ParaDiS* in short that allows such functionality. As its name suggests, ParaDiS is inspired by slimmable neural networks framework (Yu et al., 2019), where the full model variants (or so-called *switches*) consist of the model sub-networks of different widths, and all variants are trained jointly while sharing the parameters. For example, the switch denoted as $0.25\times$ is a sub-network with each layer composed of the first $25\%$ of the channels of the full network. Similarly, in ParaDiS framework we consider several variants or switches. However, ParaDiS switches include configurations distributable on several devices as illustrated on Figure 2. As such, we represent each configuration or switch as a list of widths. For example, $[0.5, 0.5]\times$ denotes a configuration consisting of two parallel networks extracted from the first and the second halves of channels of the full network, respectively. All ParaDiS model switches are trained jointly, while strongly sharing most of their parameters. Once a set of available devices is known, a suitable configuration may be selected and instantly deployed. Moreover, if a copy of the full model is available on each device, the configuration may be changed instantly without re-loading a new model with a new configuration. For example, if one device has got more computational resources for some reason (e.g., some other process has finished) configuration $[0.5, 0.25, 0.25]\times$ may be instantly changed to $[0.5, 0.5]\times$ by simply changing the sub-models executed on each device. Finally, similar to slimmable framework, ParaDiS framework is applicable for most modern convolutional neural network (CNN) architectures and for different tasks (e.g., classification, detection, identification, image restoration and super-resolution).

---

[1]The term "near-optimally" means here that each distributed configuration makes profit of all available resources and that it performs on par with a similar configuration trained specifically for this setup.

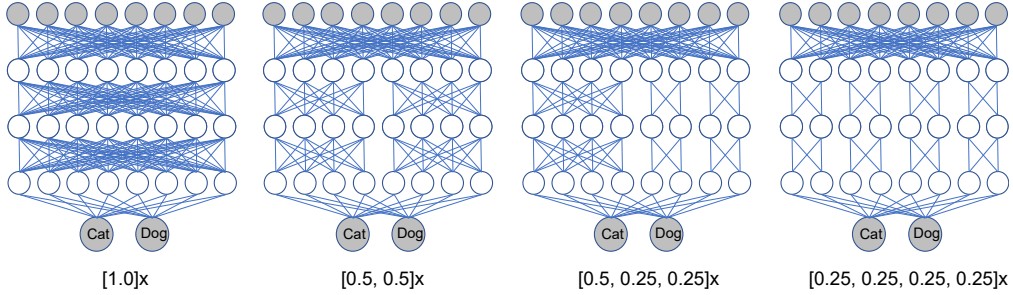

Figure 2: Illustration of ParaDiS model consisting of 4 switches. While white circles denote the neurons, gray circles denote either network inputs (e.g., image pixels) or its outputs. All the parameters are shared between switches (except for batch normalization statistics) and trained jointly.

It has been noticed in (Yu et al., 2019) that slimmable network switches may share all the parameters except the global statistics of batch normalization (BN) blocks. Individual BN statistics are adopted instead, which does not lead to any serious drawback. Indeed, individual BN statistics may be computed within a simple calibration phase after the model is trained (Yu & Huang, 2019), and they represent a very tiny overhead in terms of the total number of parameters. Since ParaDiS is an extension of the slimmable framework, we adopt in this work the same strategy of individual BN statistics. As for training, while it is inspired by the training procedure for universally slimmable (US) networks (Yu & Huang, 2019), where all switches are trained jointly, the full model is trained from the data and all other switches are distilled via knowledge distillation (KD) (Hinton et al., 2015) from the full model. We introduce two important differences. First, instead of distilling the knowledge from the full model (switch $[1.0]\times$) we distill it from a wider model (e.g., switch $[1.2]\times$) called simply *wide model* hereafter. Second, we change the KD procedure by distilling the feature activation maps before the final fully connected layer in addition to the output predicted by the wide model. We have shown that these new ingredients are important for training ParaDiS.

We investigate the proposed ParaDiS framework on ImageNet classification task using MobileNet v1 (Howard et al., 2017) and ResNet-50 (He et al., 2016) as underlying architectures and on image super-resolution task using WDSR architecture (Yu et al., 2020a). We compare ParaDiS models with the corresponding parallelly distributed configurations trained individually and with the US models (Yu & Huang, 2019) that are not distributable. First, we find that ParaDiS model performs as good as and in many cases better than the individually trained distributed configurations. Second, we observe that distributable ParaDiS switches perform almost as good as non-distributable US model switches of the same overall complexity. Third, we show that, once distributed overs several devices, ParaDiS outperforms greatly the US models. Finally, we conduct an exhaustive ablation study to show the importance of knowledge distillation, using a wide model and distilling activations.

## 2 RELATED WORK

**Compressed and flexible models.** Model compression schemes (Cheng et al., 2018) (e.g., pruning (LeCun et al., 1990) or knowledge distillation (Hinton et al., 2015)) allow satisfying particular memory and processing constraints, though they do only allow producing a fixed model for every particular setting. To go further, a new family of so-called flexible approaches allowing to instantly trade off between the accuracy and efficiency was introduced. These approaches include among others early-exit models (Huang et al., 2018), slimmable neural networks (Yu et al., 2019; Yu & Huang, 2019), Once-for-All (OFA) networks (Cai et al., 2019; Sahni et al., 2021), and neural mixture models (Ruiz & Verbeek, 2020). Moreover, flexible models like OFA networks have been shown to be useful for efficient neural architecture search (Yu et al., 2020b) and for developing dynamic inference approaches (Li et al., 2021). However, unlike our approach, all these approaches are within one device non-distributed settings.

**Sequentially distributed models.** In Neurosurgeon (Kang et al., 2017), the authors propose a dynamic partitioning scheme of an existing deep neural network between a client (end-user device,

for example) and a distant server (edge or cloud) (see also Fig. 1 (B)). The neural network can only be split between layers to form a head and a tail and the optimal partition is obtained by estimating the execution time of each part on the different devices. The main drawback of this approach is the added communication time occurring between the different partitions. To this end other optimization strategies have been developed and intermediate feature compression schemes have been proposed (Matsubara et al., 2021). However, as we have discussed above, distributing a model sequentially does not allow making use of a plurality of devices to reduce processing latency.

**Parallelly distributed models.** In contrast to sequential distribution, distributing model parallelly with no communication (see Fig. 1 (D)) allows reducing processing latency without increasing communication burden at the same time. Several recently proposed approaches (Bhardwaj et al., 2019; Asif et al., 2019) suggest creating such efficient parallel models via knowledge transfer from a bigger single pre-trained model. Wang et al. (2020) revisit model ensembles that are parallellizable as well, and show that they are almost as efficient as individual non-distributable models of the same complexity. However, in contrast to our proposal, none of those models are sharing parameters.

## 3 ParaDiS neural networks

### 3.1 Switchable batch normalization statistics

Batch normalization (BN) (Ioffe & Szegedy, 2015) is an essential block present nowadays in most popular deep CNNs. BN is very simple and consists in normalizing CNN features in each layer and channel with mean and variance of the current mini-batch. While during training the statistics of the current mini-batch are considered, during testing these statistics are replaced by global statistics computed as moving averages over the whole or a part of the training set. Because of this inconsistency between training and testing, the BN parameters cannot be simply shared in slimmable neural networks (Yu et al., 2019; Yu & Huang, 2019). As such, it was first proposed in (Yu et al., 2019) to consider a so-called *switchable* BN block having separate set of parameters for every switch. It was then shown that it is sufficient to consider only switchable BN statistics, while the remaining parameters of the BN block might be shared between switches. Finally, it was proposed (Yu & Huang, 2019) to compute switchable BN statistics in a so-called *calibration phase* after training. For each switch the calibration consists simply in a forward pass over the corresponding network over a subset of the training set, so as to compute global BN statistics. Since the proposed ParaDiS framework is based on slimmable networks, we here adopt switchable BN statistics together with the calibration procedure.

### 3.2 Training ParaDiS networks

The switches of a ParaDiS network are trained jointly using a training procedure inspired by US networks training (Yu & Huang, 2019), though with several modifications we introduced that we have shown to be important for achieving good performance. Recall that US networks training (Yu & Huang, 2019) consists in updating all switches jointly (though sampling a limited number of switches per epoch), while only the full $1.0\times$ model is trained from data and all other switches are trained from the output of the full model via a knowledge distillation (KD) (Hinton et al., 2015). This procedure is referred to as *inplace knowledge distillation (IPKD)* since all the switches are strongly sharing their parameters. More specifically, assuming here without loss of generality an image classification task, the full $1.0\times$ model in US training (Yu & Huang, 2019) is updated based on the true labels using the conventional cross-entropy (CE) loss

$$loss = \mathcal{L}_{CE}(y', y) = \frac{1}{C} \sum\nolimits_{c=1}^{C} y_c \log(y'_c), \tag{1}$$

where $C$ is the number of classes, and $y$ and $y'$ are $C$-dimensional hot vectors of true and predicted labels, respectively. All other switches are trained by optimizing the following KD loss [2]:

$$loss = \mathcal{L}_{CE}(\hat{y}, y'), \tag{2}$$

---

[2]As compared to Eq. (2), more sophisticated KD-losses are often considered (Hinton et al., 2015; Ozerov & Duong, 2021), including those that take into account the true labels $y$ as well. However, we keep here this simple formulation since it was used in US training (Yu & Huang, 2019).

where $\hat{y}$ and $y'$ are hot vectors of labels predicted by the current switch and by the full model, respectively. We adopt this approach to train the ParaDiS networks, while improving it with two novel add-ons (that are important for training) explained just below and depicted on Fig. 3.

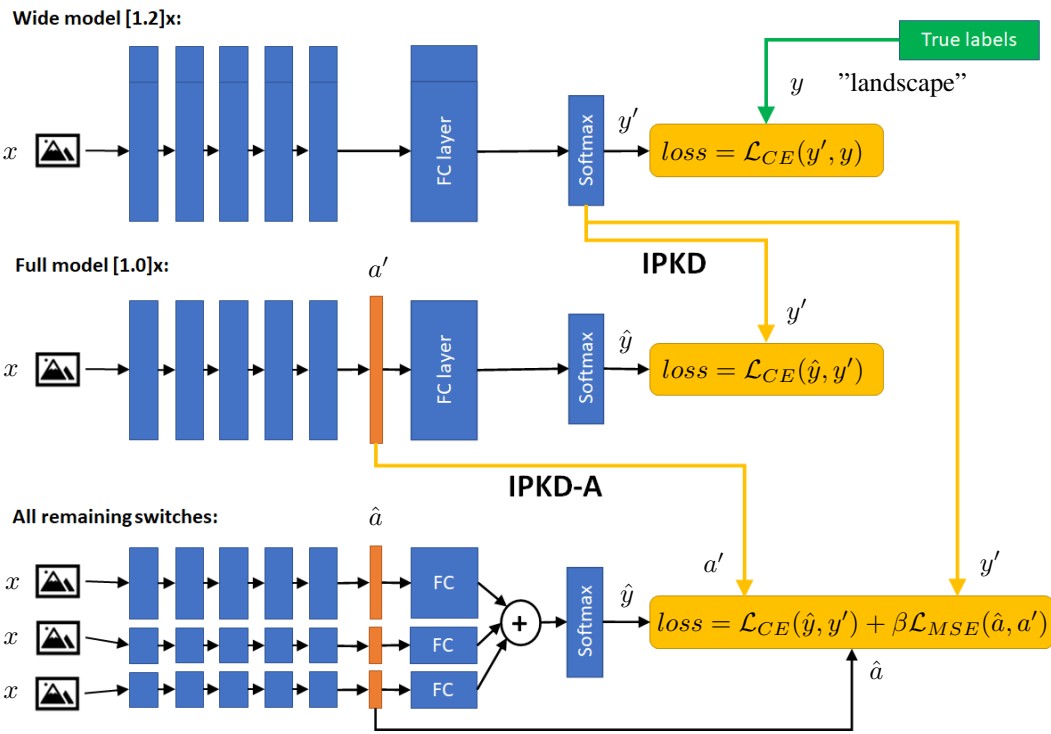

Figure 3: Visualization of the ParaDiS network training scheme, also described by Alg. 1.

**Knowledge distillation from a wider model.** Within our early experiments on training ParaDiS MobileNet v1 network with 4 switches (as shown on Fig. 2) on ImageNet dataset we tried applying directly the US training framework, but this did not lead to a satisfactory result. Indeed, the full $[1.0]\times$ model was not training well enough under-performing by about 1 % the individual MobileNet v1. Since all the other switches were distilled from the full model, their performances were degraded by at least the same amount, as compared to the expected ones. We believed that this is because all the switches span the full $1.0\times$ network width, while sharing their parameters, and thus the full $[1.0]\times$ model is not "free enough" to train well. In fact, this is not the case for slimmable (Yu et al., 2019) or US (Yu & Huang, 2019) networks, where all other switches have smaller widths as compared to the full model. This gave us the idea to perform IPKD from a wider (e.g., $[1.2]\times$) network, which is simply called hereafter *wide network*. Indeed, the wide network is always wider than all other ParaDiS switches, and thus it is not constrained and can train better. Note also that the wide network is only needed for training, as such its additional parameters (as compared to the full network) may be simply discarded after training. Moreover, as another alternative, any other big network (completely independent from the ParaDiS network) may be also used as wide network.

**Distilling feature activation maps.** Another issue we have noticed is that the training of ParaDiS network distributed switches is not always very stable. Indeed, since the fusion occurs only after the last layer (see, e.g., Fig. 3, bottom) training of one sub-network depends on the outputs of all other sub-networks, and thus not very "independent". To make this IPKD training more "sub-model dependent", we modify the IPKD to distill as well the full $[1.0]\times$ model feature activation maps just before the final fully connected (FC) layer to all other sub-models. More precisely, we propose to modify the KD-loss (2) as follows:

$$loss = \mathcal{L}_{CE}(\hat{y}, y') + \beta\mathcal{L}_{MSE}(\hat{a}, a'), \qquad \mathcal{L}_{MSE}(\hat{a}, a') = \frac{1}{N}\|\hat{a} - a'\|_2^2, \qquad (3)$$

where $\beta$ is a constant parameter, $a'$ is an $N$-dimensional vector containing a vectorization of the feature activation maps before the final FC layer, and $\hat{a}$ is the vector including the respective activation maps of the corresponding switch [3] (see Fig. 3). We believe that this should stabilize training since when $\beta$ grows training of each sub-network becomes more and more independent of the outputs of other sub-networks within a switch. Such kind of regularisation of KD-loss has been already considered in (Bhardwaj et al., 2019) (see Eq. (6) in the paper), but not in the case of inplace knowledge distillation as we consider here. We refer to this proposed activation distillation as *IPKD-A*.

Note also that in contrast to US training (Yu & Huang, 2019), where a random selection of switches is updated at each iteration, we update all the switches at each iteration. The resulting procedure is schematized on Figure 3 and described in detail by Algorithm 1.

---

**Algorithm 1:** Training ParaDiS network $M$ (see also Fig. 3).

---

**Require:** Define *list of switches*, e.g., as on Fig. 2:
$$[[1.0]\times, [0.5, 0.5]\times, \ldots, [0.25, 0.25, 0.25, 0.25]\times]$$
**Require:** Define *wide model*, e.g., as $[1.2]\times$ switch.

1 Initialize training settings of the main shared network $M$.
2 **for** $t = 1, \ldots, T_{iters}$ **do**
3      Get next mini-batch of data $x$ and label $y$.
4      Clear gradients, $optimizer.zerograd()$.
5      Execute wide model, $y' = M(x, wide\ model)$.
6      Compute loss, $loss = \mathcal{L}_{CE}(y', y)$ (Eq. (1)).
7      Accumulate gradients, $loss.backward()$.
8      Stop gradients of $y'$ as label, $y' = y'.detach()$.
9      Execute full model, $(\hat{y}, a') = M(x, [1.0]\times)$, where $a'$ is the activation maps vector.
10      Compute loss, $loss = \mathcal{L}_{CE}(\hat{y}, y')$ (Eq. (2)).
11      Accumulate gradients, $loss.backward()$.
12      Stop gradients of $a'$ as activation, $a' = a'.detach()$.
13      **for** *switch in list of switches* **do**
14          **if** *switch !=* $[1.0]\times$ **then**
15              Execute current switch, $(\hat{y}, \hat{a}) = M(x, switch)$, where $\hat{a}$ is the activation maps vector..
16              Compute loss, $loss = \mathcal{L}_{CE}(\hat{y}, y') + \beta\mathcal{L}_{MSE}(\hat{a}, a')$ (Eq. (3)).
17              Accumulate gradients, $loss.backward()$.
18      Update weights, $optimizer.step()$.

---

## 4 EXPERIMENTS

We evaluate our approach on the ImageNet classification dataset (Deng et al., 2009) with 1000 classes. We conducted experiments on one light-weight model (MobileNet v1) and one large model (ResNet-50). For both models 4 different switches are trained at each epoch: $[4 \times 0.25]\times$ [4], $[0.5, 0.5]\times$, $[1.0]\times$ and $[1.2]\times$. We compare our performance with US model and individually trained models ($[4 \times 0.25]\times$, $[0.5, 0.5]\times$, $[0.5, 0.25, 0.25]\times$ and $[1.0]\times$). Note that for a fair comparison the reference US model, similarly to our ParaDiS model, was trained while distilling from a wide $[1.2]\times$ model instead of the full $[1.0]\times$ model as in the original work (Yu & Huang, 2019). The corresponding approach is then referred to as *US Wide* (some results on the effect of wide distillation for US and slimmable Nets are in Appendix B).

We evaluate the accuracy (1 - top-1 error) on the center $224 \times 224$ crop of images in the validation set. For all models, we show the complexity in terms of Millions of Multiply-Adds (MFLOPs). For readability the results of wide $[1.2]\times$ model as well as of small switches for the reference models are not shown on the plots, however detailed results can be found in the corresponding appendices.

---

[3]Since a switch of a ParaDiS network does not necessary span the range of the full model (e.g., it might be $[0.25, 0.25]$), vector $\hat{a}$ in (3) may be shorter than $a'$. In this case we assume that $\hat{a}$ is simply padded by zeros to have the same length.

[4]$[4 \times 0.25]\times$ is a shorter notation for $[0.25, 0.25, 0.25, 0.25]\times$.

However, this representation in terms of total MFLOPs does not represent any gain of ParaDiS over US and Slimmable models in terms of latency. Also, it is not trivial since such gain depends on the available devices and their capacities. To represent this gain, we add additional curves to the main graphs as follows. We assume a situation when all the devices have equal capacities, and we have any number of them. In that case the US switch $0.5\times$ and the ParaDiS switch $[0.5, 0.5]\times$ will run with the same latency (neglecting the communication latency that is small) and the same number of MFLOPs per device, while $[0.5, 0.5]\times$ has a superior accuracy. In summary, we add to the corresponding graphs below a curve tagged as "distributed" that shows the accuracies of switches $[0.25, 0.25, 0.25, 0.25]\times$, $[0.5, 0.5]\times$, $[1.0]\times$ as a function of MFLOPs for one sub-model (and not a sum of MFLOPs for all sub-models). This curve is representative of a deployment with several equal capacities devices and shows the real advantage of ParaDiS over US and Slimmable models.

In order to diversify the tasks and architectures, we also investigated ParaDiS for image super-resolution using WDSR model (Yu et al., 2020a). Detailed results can be found Appendix D, they are inline with those for ImageNet classification.

## 4.1 MobileNet v1

We use the default training and testing settings (Howard et al., 2017) for MobileNet v1 with the same adjustments as proposed in US-Nets paper (Yu & Huang, 2019): 250 epochs, stochastic gradient descent as optimizer (instead of RMSProp), a linearly decaying learning rate from 0.5 to 0 and a batch size of 1024 images.

The results are shown on Figure 4 (see Appendix C for details). First, one may see that ParaDiS performs on par with or in some cases ($[4\times0.25]\times$ and $[0.5, 0.5]\times$) better than the individual models. This is inline with similar findings for slimmable and US models (Yu et al., 2019; Yu & Huang, 2019), and this is mostly thanks to implicit KD (parameters sharing) and explicit KD. Second, yet more striking is the fact that the distributable ParaDiS switches perform as good as non-distributable US switches of the same overall complexity. Finally, while $[0.5, 0.25, 0.25]\times$ switch has not been explicitly trained at all, it performs well. This is possibly because it spans the full width, as other switches, and all its sub-models were trained within other switches. Thanks to the "distributed" curve we see also a great improvement of distributed ParaDiS over US model.

## 4.2 ResNet-50

For ResNet-50, we use the same parameters as in slimmable networks (Yu et al., 2019): 100 epochs with a batch size of 256 images, a learning rate of 0.1 divided by 10 at 30, 60 and 90 epochs. We use stochastic gradient descent (SGD) as optimizer, Nesterov momentum with a momentum weight of 0.9 without dampening, and a weight decay of 1e-4 for all training settings.

The results are shown on Figure 5 (see Appendix C for details), where we also compare with slimmable and slimmable wide, since they perform better than US wide for ResNet-50. Almost the same conclusions as for MobileNet v1 may be drawn with the following differences. First, the improvement of ParaDiS over individual models is yet more significant. Second, the performance of ParaDiS $[4 \times 0.25]\times$ switch drops a little as compare to the similar US switch.

## 5 Ablation Study

In this section we show the impact of different training settings such as Inplace Knowledge Distillation (IPKD), the use of a wide network for the IPKD and the importance of distilling the activation. We use MobileNet v1 and Resnet-50 trained on ImageNet to measure the contribution of these different parameters (additional plots may be found in Appendix E).

First, results in Table 1 indicate the importance of IPKD and of the wide model for ParaDiS training. At the same time it shows that distilling activations (either with $\beta = 0.1$ or $\beta = 1$) does not help for training ParaDiS MobileNet v1. However, results in Table 2 confirm the importance of distilling activations for training ParaDiS ResNet-50.

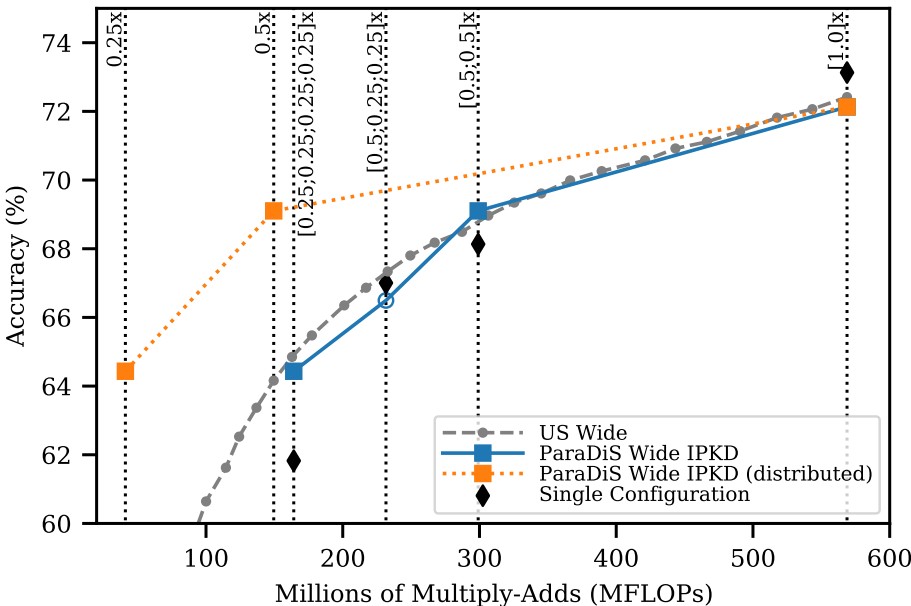

Figure 4: Complexity in MFLOPs vs accuracy of MobileNet v1 trained on Imagenet for different switches (US-Nets and ParaDiS). The vertical dotted lines indicate the complexity of specific switches. The diamond points are individual models trained for a particular switch. The square blue points are switches explicitly trained, while the empty blue circle is never trained, only tested.

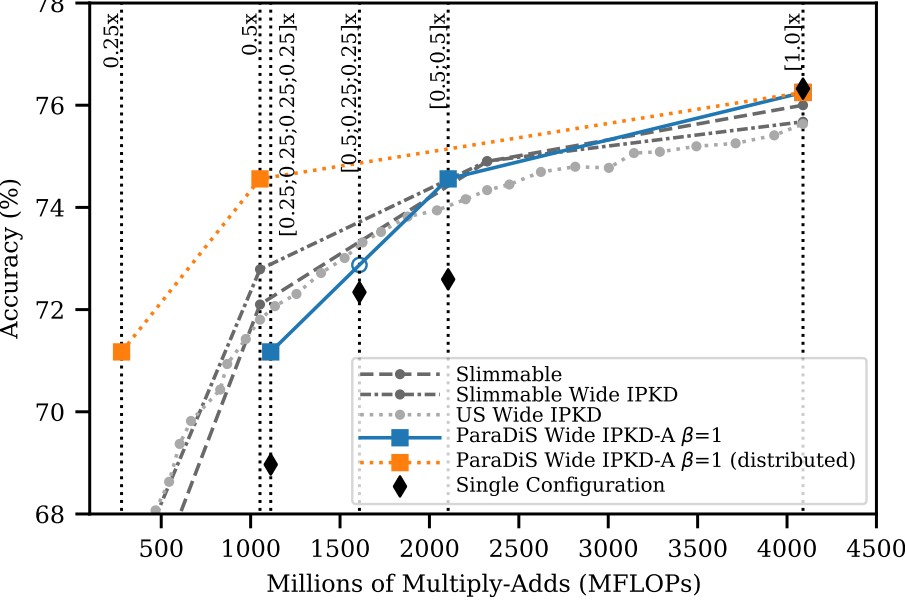

Figure 5: Complexity in MFLOPs vs accuracy of ResNet-50 trained on Imagenet for different switches (US-Nets, Slimmable and ParaDiS). The vertical dotted lines indicate the complexity of specific switches. The diamond points are individual models trained for a particular switch. The square blue points are switches explicitly trained, while the empty blue circle is never trained, only tested.

| | $[1.2]\times$ | $[1.0]\times$ | $[0.5, 0.5]\times$ | $[0.5, 0.25, 0.25]\times$ | $[4 \times 0.25]\times$ |
|---|---|---|---|---|---|
| ParaDiS Wide IPKD-A $\beta$=1 | 72.64% | 71.77% | 68.23% | 66.32% | 63.66% |
| ParaDiS Wide IPKD-A $\beta$=0.1 | 73.04% | 71.99% | 68.59% | 66.36% | 64.18% |
| ParaDiS Wide IPKD | 73.18% | 72.13% | 69.10% | 66.49% | 64.43% |
| ParaDiS IPKD | N/A | 71.52% | 68.21% | 66.10% | 63.51% |
| ParaDiS | N/A | 70.83% | 67.67% | 64.41% | 63.28% |

Table 1: Impact of different hyperparameters on the training of ParaDiS (MobileNet v1 + ImageNet).

| | $[1.2]\times$ | $[1.0]\times$ | $[0.5; 0.5]\times$ | $[0.5; 0.25; 0.25]\times$ | $[4 \times 0.25]\times$ |
|---|---|---|---|---|---|
| ParaDiS Wide IPKD-A $\beta = 1$ | 76.48% | 76.25% | 74.56% | 72.87% | 71.17% |
| ParaDiS Wide IPKD | 76.48% | 76.06% | 74.38% | 73.02% | 71.34% |

Table 2: Impact of the distillation of the activation when training Resnet-50 for ImageNet.

## 6   Scalability issues

Within the proposed ParaDiS networks there is potentially many more possible switches than in slimmable or US nets. Indeed, each ParaDiS switch sub-network might be of any width. As such, it seems quite important to be able to scale the approach to more switches than the 4 used in our experiments.

We have shown in our experiments that some switches might perform satisfactory though they were never explicitly trained, e.g., switch $[0.5, 0.25, 0.25]\times$ (see Figs. 4 and 5), possibly because it spans the full width, as other switches, and all its sub-models were trained within other switches. Based on this observation, we conjecture that if we have smaller sub-models, e.g., by considering wider full model, we may have more non-trained satisfactory switches "for free". For example, if we split the full model in 8 sub-models and train $[1.2]\times$, $[1.0]\times$, $[0.5, 0.5]\times$, $[4 \times 0.25]\times$ and $[8 \times 0.125]\times$, we would have 6 other switches coming for free. We show experimentally in Appendix F.1 that this path to scale up ParaDiS framework is viable.

Alternatively, we have also tried training ParaDiS with 8 switches, where we also vary the width spanned by the switch. The results we obtained (see Appendix F.2 for details) are very unsatisfactory: the accuracy of each switch is at least 4 % below the accuracy of the corresponding US net switch. What remains unclear and should be a subject of a further study is: *Is this performance drop because all these switches cannot co-exist within the shared ParaDiS parameters setting or because we have not found yet how to train them correctly?*

## 7   Conclusion

In this work we have introduced parallelly distributable slimmable (ParaDiS) neural networks. These networks are splittable in parallel between various device configurations without retraining. While so-called flexible networks allowing instant adjusting to a given complexity have been already widely studied, the kind of "flexibility-to-distribute" property of ParaDiS networks is considered for the first time to our best knowledge. ParaDiS networks are inspired by one-device flexible slimmable networks and consist of several distributable configurations or switches that strongly share the parameters between them. We evaluated ParaDiS framework on MobileNet v1 and ResNet-50 architectures on ImageNet classification task and on WDSR architecture for image super-resolution task. First, we have shown that ParaDiS models perform as good as and in many cases better than the individually trained distributed configurations. Second, we have found that distributable ParaDiS switches perform almost as good as non-distributable universally slimmable model switches of the same overall complexity. Finally, we have shown that once distributed over several devices, ParaDiS outperforms greatly slimmable models.

A very important topic for further study would be scaling the ParaDiS networks to more configurations/switches than 4. As we have demonstrated in Appendix F.1, one way to achieve that consists in considering more sub-models at the last level of split with "for freee"non-trained switches as a result. To yet improve on that, one might also consider width-overlapping sub-models.

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

# A    PARADIS AND SLIMMABLE NEURAL NETWORKS REPRESENTATION

**Slimmable Neural Nets**

**ParaDiS Neural Nets**

Figure 6: A graphical representation of Slimmable and ParaDiS switches. For each switch, the surface of the blue area is roughly equal to the relative number of parameters, as compared to the full $1.0\times$ model, and the relative number of FLOPs.

Here we present a graphical representation of Slimmable and ParaDiS switches that allows a better understanding of these models. An important observation is that in most cases slimming a CNN from $1.0\times$ (full model) to $W\times$ leads to roughly $W^2$ times more parameters in switch $1.0\times$ than in switch $W\times$ (the same is valid for the computational complexity in FLOPs). This is because most parameters reside in the weights of convolutional and fully connected layers, and the number of these weights scales with network width $W$ as $W^2 const$. The same holds for ParaDiS sub-models. As such we have found very intuitive to represent slimmable switches as squares and ParaDiS switches as sets of squares, as shown on Figure 6. For each switch within this representation the surface of the blue area is roughly equal to the relative number of parameters, as compared to the full $1.0\times$ model, and the relative number of FLOPs. For example, from this representation we may conclude that ParaDiS switch $[4 \times 0.25]\times$ has roughly the same complexity as slimmable or US switch $0.5\times$ since their surfaces are equal. This is confirmed by the fact that, as we may see from Tables 4 and 5 below, the US switch that is closest in complexity to ParaDiS switch $[4 \times 0.25]\times$ is of width $0.525 \approx 0.5$. Finally, Yu et al. (2020b) were also using such a representation with "squares" for slimmable networks (see Fig. 1 in (Yu et al., 2020b)).

# B    WIDE VS. NON-WIDE FOR US AND SLIMMABLE NETS

Table 3 regroups the results of experiments using US or slimmable networks for MobileNet v1 or Resnet-50. Interestingly, it is MobileNetv1 that benefits the most of the wide network, the results for slimmable Resnet-50 show a slight loss of performance for the biggest switches and a small gain for the lowest switches.

# C    DETAILED RESULTS FOR IMAGENET CLASSIFICATION

Tables 4 and 5 summarize the detailed results corresponding to those plotted on Figures 4 and 5, respectively. We may see in Table 5 that for most of ParaDiS switches we do not indicate results of slimmable networks. This is simply because the complexities of slimmable switches do not correspond exactly to those of ParaDiS switches. As for results of US nets on both Tables 4 and 5,

|  |  | (1.2×) | (1.0×) | (0.75×) | (0.5×) | (0.25×) |
|---|---|---|---|---|---|---|
| MobileNet v1 | US Wide IPKD | 72.98% | 72.42% | 69.34% | 64.16% | 55.02% |
|  | US IPKD | N/A | 71.80% | 69.50% | 64.20% | 55.70% |
| ResNet-50 | US Wide IPKD | 75.85% | 75.63% | 74.34% | 71.80% | 66.41% |
|  | US IPKD | N/A | 75.77% | 74.52% | 72.22% | 67.18% |
|  | Slimmable Wide IPKD | 75.89% | 75.68% | 74.91% | 72.79% | 66.37% |
|  | Slimmable | N/A | 76.00% | 74.90% | 72.10% | 65.00% |

Table 3: Impact of using a wider configuration on the training of US Nets (MobileNet v1 + ImageNet) and US/slimmable Nets (ResNet-50 + ImageNet).

for each ParaDiS switch we indicate the accuracy of the switch having the closest complexity to the corresponding ParaDiS switch.

| ParaDiS switch (corresponding US switch) | $[1.2]\times$ (1.2×) | $[1.0]\times$ (1.0×) | $[0.5; 0.5]\times$ (0.725×) | $[0.5; 0.25; 0.25]\times$ (0.625×) | $[4 \times 0.25]\times$ (0.525×) |
|---|---|---|---|---|---|
| ParaDiS Wide IPKD | 73.18% | 72.13% | 69.10% | 66.49% | 64.43% |
| US Wide IPKD | 72.98% | 72.42% | 68.97% | 67.34% | 64.85% |
| Individual Models | N/A | 73.13% | 68.13% | 67.00% | 61.83% |

Table 4: Accuracy comparison for different switches of MobileNet v1. This table summarizes more in details the results plotted on Figure 4. As for US results, the accuracy of the switch having the closest complexity to the corresponding ParaDiS switch is given every time.

| ParaDiS switch (corresponding US switch) | $[1.2]\times$ (1.2×) | $[1.0]\times$ (1.0×) | $[0.5; 0.5]\times$ (0.7×) | $[0.5; 0.25; 0.25]\times$ (0.625×) | $[4 \times 0.25]\times$ (0.525×) |
|---|---|---|---|---|---|
| ParaDiS Wide IPKD-A, $\beta = 1$ | 76.48% | 76.25% | 74.56% | 72.87% | 71.17% |
| US Wide IPKD | 75.85% | 75.63% | 74.16% | 73.31% | 72.07% |
| Slimmable Wide IPKD | 75.89% | 75.68% | N/A | N/A | N/A |
| Slimmable | N/A | 76.00% | N/A | N/A | N/A |
| Individual Models | N/A | 76.32% | 72.59% | 72.34% | 68.97% |

Table 5: Accuracy comparison for different switches of ResNet-50. This table summarizes more in details the results plotted on Figure 5. As for US results, the accuracy of the switch having the closest complexity to the corresponding ParaDiS switch is given every time.

## D   DETAILED RESULTS FOR IMAGE SUPER-RESOLUTION

In order to investigate ParaDiS on a different task and with yet different architecture, we study it here for image super-resolution task using WDSR architecture (Yu et al., 2020a). Inspired by (Yu & Huang, 2019), we experiment with DIV2K dataset (Timofte et al., 2017) which contains 800 training and 100 validation 2K-resolution images, on the task of bicubic ×2 image super-resolution.WDSR network has no batch normalization layer, instead weight normalization (Salimans & Kingma, 2016) is used. We use the latest implementation of the WDSR network released in April 2020 [5] and adapt the model and training to support Slimmable, Universally Slimmable (US) and ParaDiS configurations. Note that, to our best knowledge, there is no publicly available implementation of WDSR US-Nets as reported in (Yu & Huang, 2019). We use the same maximum width (64 channels), number of residual blocks (8) and width multiplier (4) as reported by Yu & Huang (2019). The default hyper-parameters (number of training epochs, learning rate, weight decay ...) are used for all experiments.

We report results in Figure 7 and show that ParaDiS exhibits a performance on par with US-Nets and slightly below individually trained models. When distributing the ParaDiS network on several devices (orange dotted curve in Figure 7), the benefits of the parallelization allows our model to outperform individual models running on a single device. The results are consistent with reported

---

[5]https://github.com/ychfan/wdsr

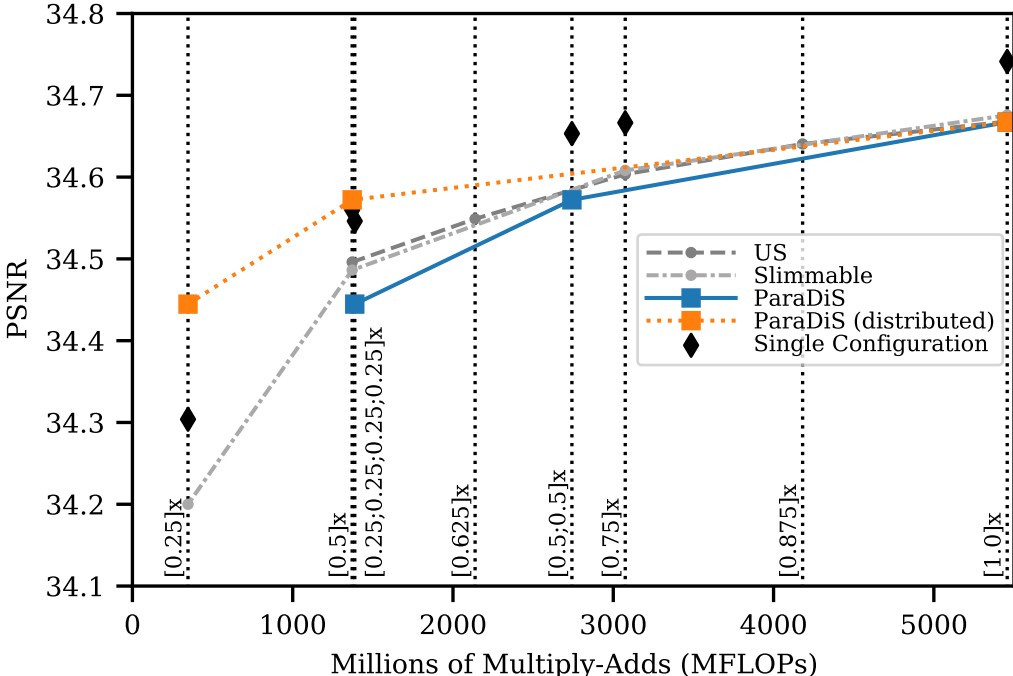

Figure 7: Complexity in MFLOPs vs PSNR of different WDSR networks (US-Nets, Slimmable and ParaDiS) trained on DIV2K on the task of bicubic ×2 image super-resolution. The vertical dotted lines indicate the complexity of specific switches. The diamond points are individual models trained for a particular switch.

results for individual models (Yu et al., 2020a) and for US-Nets without inplace distillation (Yu & Huang, 2019). However we were unable to reproduce the results reported for US-Nets when inplace distillation is used during training. All models showed a degradation of performance when using knowledge distillation.

# E    ABLATION STUDY

In this section we present the results of our ablation study for MobileNet v1 and Resnet-50 trained on ImageNet.

## E.1    WIDE VS. NON-WIDE AND IPKD VS. NO KD

First we study the impact of IPKD and distilling from a wide network. Figure 8 clearly shows the benefit of IPKD from a wide model by comparing the ParaDiS models trained without Knowledge Distillation, trained with IPKD from model $[1.0]\times$ and trained with IPKD from wide model $[1.2]\times$. It is interesting to note a positive impact of using Knowledge Distillation on untrained switches. In our case we see a big improvement in performance, thanks to IPKD, for the $[0.5, 0.25, 0.25]\times$ switch that is never explicitly trained.

## E.2    DISTILLATION OF THE ACTIVATIONS

We present in this section the results of experiments using MobileNet v1 and Resnet-50 with and without distilling feature activation maps from the $[1.0]\times$ model.

From Figure 9 one may notice that the distillation of feature activation maps does not help training different switches of ParaDiS MobileNet v1.

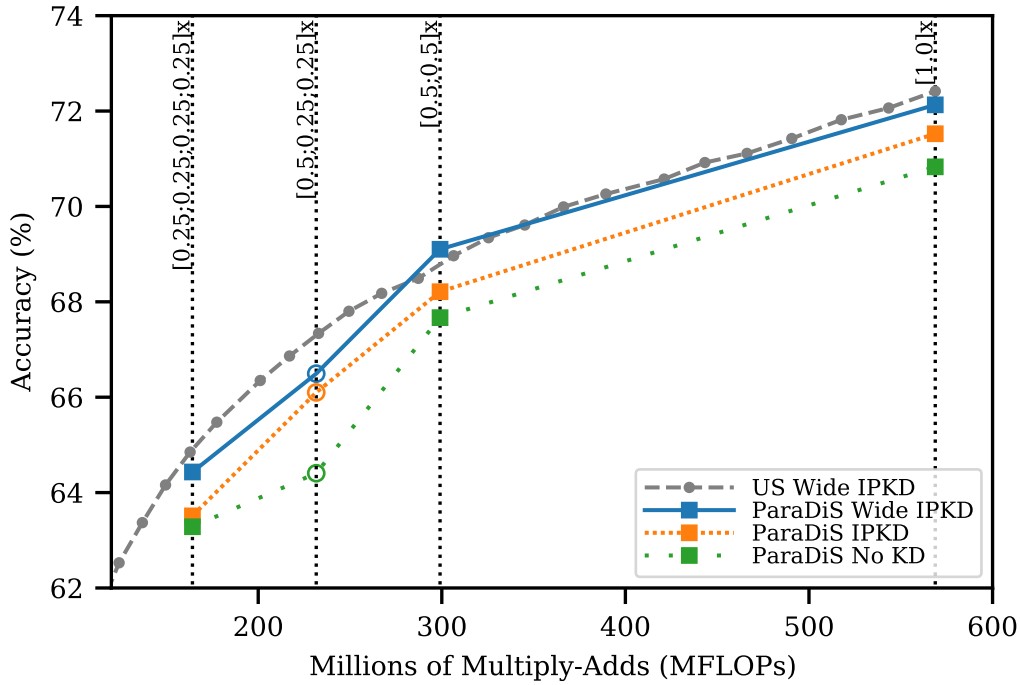

Figure 8: Impact of of IPKD and distilling from a wide network for MobileNet v1. These results correspond to 3 rows from Table 1.

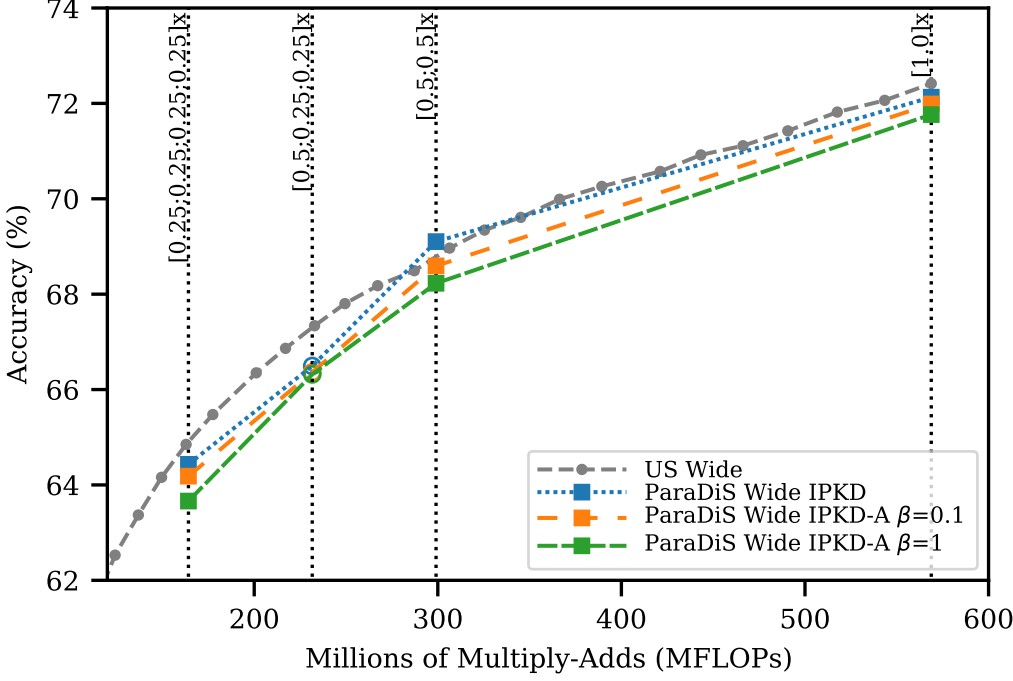

Figure 9: Impact of distilling the activation for MobileNet v1. These results correspond to 3 rows from Table 1.

From Figure 10 one may see that the benefit of distillation of feature activation maps is clearly visible especially for the largest switches of ParaDiS ResNet-50.

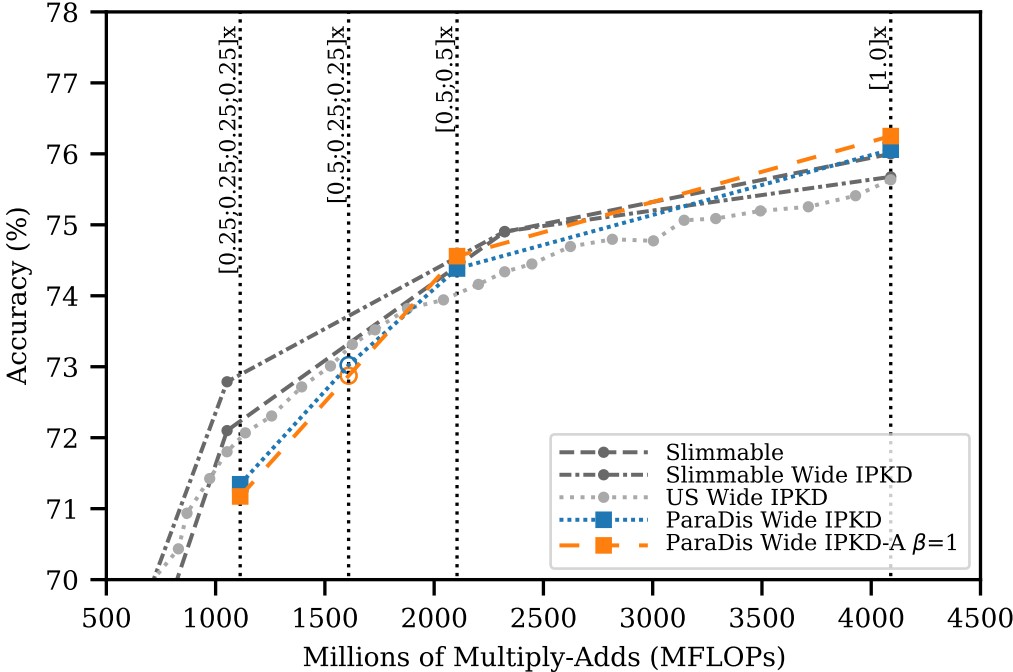

Figure 10: Impact of distilling the activation for ResNet-50. These results correspond to those from Table 2.

## F  SCALING PARADIS MODELS TO MORE SWITCHES

### F.1  NON-TRAINED SWITCHES "FOR FREE"

First, we conjecture that if we train for example switches $[1.2]\times$, $[1.0]\times$, $[0.5, 0.5]\times$, $[4 \times 0.25]\times$, and $[8 \times 0.125]\times$, we would have 6 other switches coming for free:

- $[0.5, 0.25, 0.25]\times$,
- $[0.5, 0.25, 0.125, 0.125]\times$,
- $[0.5, 0.125, 0.125, 0.125, 0.125]\times$,
- $[0.25, 0.25, 0.25, 0.125, 0.125]\times$,
- $[0.25, 0.25, 0.125, 0.125, 0.125, 0.125]\times$,
- $[0.25, 0.125, 0.125, 0.125, 0.125, 0.125, 0.125]\times$,

because they span the full width as other trained switches, and all their sub-models were trained within other switches.

Second, we have chosen ResNet-50 to verify this conjecture. Indeed, ImageNet v1 is already too optimized (thus thin) to be split into sub-models of widthes $0.125$. The results are shown on Figure 11. We may see that the performance of model trained on 5 switches drops just slightly (less than 1 % of accuracy) as compared to the one trained on 4 switches. Moreover, the performance of 6 non-trained switches coming "for free" seems to be descent, assuming especially that these switches are distributable over several devices. As such, this way of scaling ParaDiS to more switches seems to be viable.

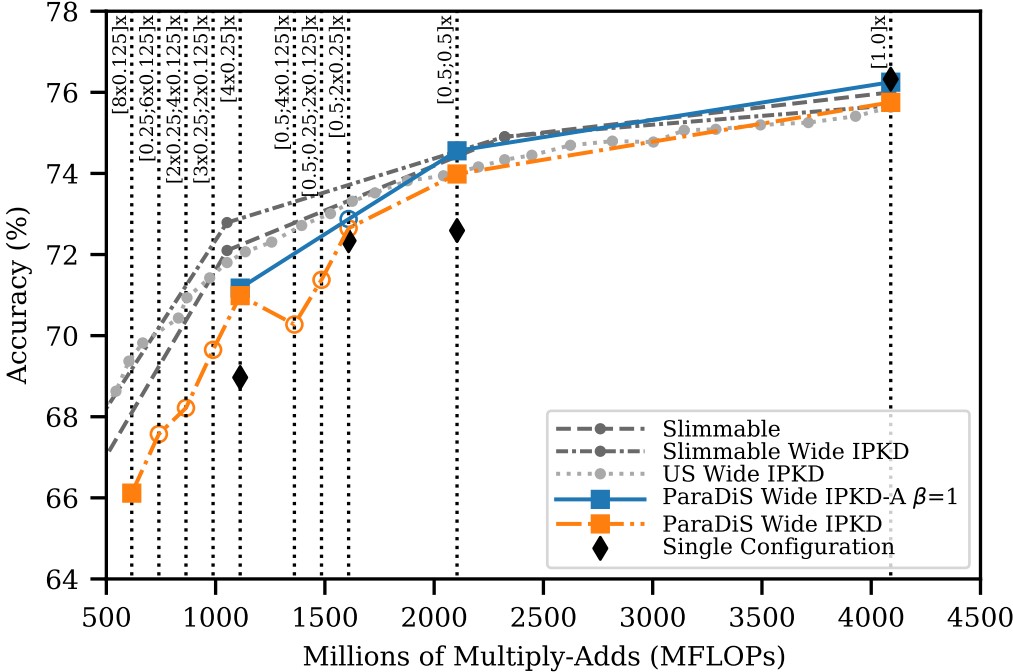

Figure 11: Complexity in Millions of FLOPs vs accuracy of ResNet-50 trained on Imagenet with 5 switches (orange curve): $[1.2]\times$, $[1.0]\times$, $[0.5, 0.5]\times$, $[4 \times 0.25]\times$, and $[8 \times 0.125]\times$, versus ResNet-50 trained on 4 switches (blue curve). The vertical dotted lines indicate the complexity of specific switches. The diamond points are individual models trained for a particular switch. The square points are switches explicitly trained, while the empty circles are never trained, only tested.

### F.2 VARYING THE WIDTH SPANNED BY THE SWITCHES

This section presents detailed results when scaling ParaDiS to more switches. We used MobileNet v1 trained on ImageNet for the experiments. 8 different switches are explicitly trained at each epoch (compared to 4 in the remainder of the experiments): $[1.2]\times$, $[1.0]\times$, $[0.5, 0.5]\times$, $[0.5] \times [4 \times 0.25]\times$, $[3 \times 0.25]\times$, $[2 \times 0.25]\times$ and $[0.25]\times$. Figure 12 shows the difference in performance between 4 and 8 switches.

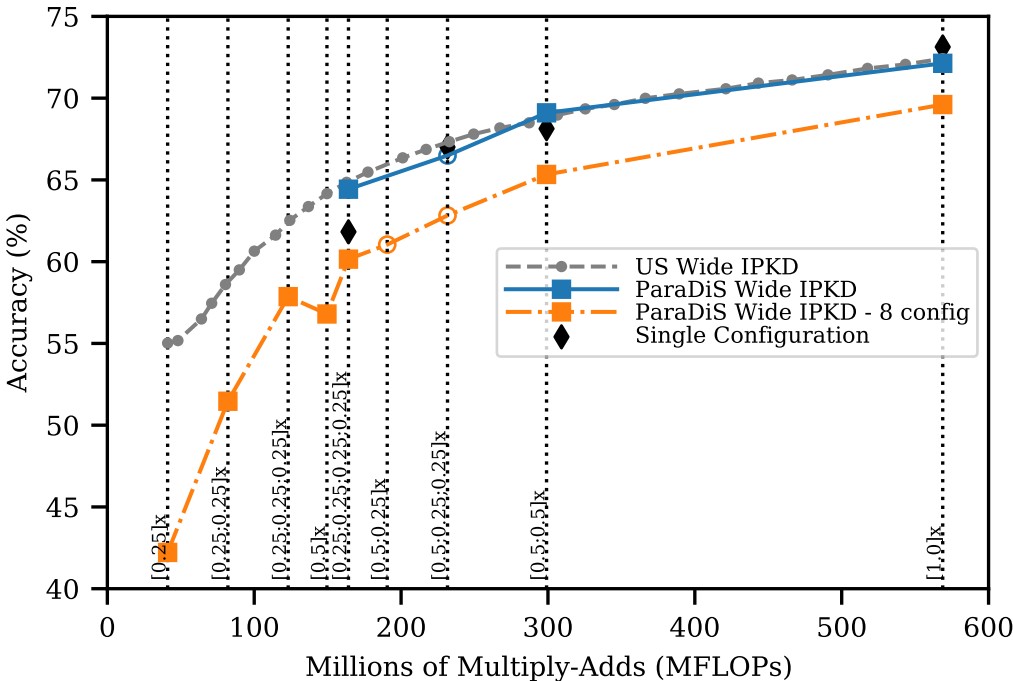

Figure 12: Complexity in Millions of FLOPs vs accuracy of MobileNetv1 trained on Imagenet with 8 switches. The vertical dotted lines indicate the complexity of specific switches. The diamond points are individual models trained for a particular switch. The square points are switches explicitly trained, while the empty circles are never trained, only tested.

