# OpenReview forum: "ParaDiS: Parallelly Distributable Slimmable Neural Networks"
_ICLR.cc/2022/Conference — ICLR 2022 Submitted_

### Official Review · Reviewer_eUCM · 2021-11-01

**Correctness:** 3
**Technical Novelty And Significance:** 2
**Empirical Novelty And Significance:** 2
**Recommendation:** 5
**Confidence:** 4

**Main Review:**


Strengths:

1. It is a practical and helpful extension to execute the slimmable networks in a parallel way, which has the potential to leverage the computation power of multiple devices.

2. The paper is well written and easy to follow.

3. Comprehensive studies are listed to verify each component in the proposed approach.

Weakness:

1. Methodologically there is no fundamental differences from slimmable networks. The paper proposes 1) distilling from wider networks and 2) distilling feature maps, both of which are not new.

2. There is little details on how to execute the parallel inference across multiple devices, which I think is a major issue of this paper. For instance:
- Does it mean there are three devices holding each part of the [0.5, 0.25, 0.25] configuration? If so, there is inevitable communication among devices.
- It would be more interesting to show the strategy to split the model across different devices, depending on their resources (memory or computation).

3. No simulation results on parallel execution of ParaDis. I may still worry about the latency overhead comparing with the inference of a singe device, especially if different devices have different computation power, or there is any communication among devices.

4. It seems the width configurations are not scalable and sensitive to width configurations as discussed in Section 6. From to Figure 1, is it necessary that all neurons in the first width configuration are fully connected (in terms of the performance)?


Detailed Comments:

1. It is still not clear to me the following statement: "Indeed, since the BN statistics are computed via a forward pass during the calibration phase, the statistics of a particular sub-model are independent of the switch it belongs to. In other words, BN statistics of the last 0.25-width sub-model will be the same, whether they are computed within [0.5, 0.25, 0.25]× switch or within [0.25, 0.25, 0.25, 0.25]× switch. To summarize, the calibrated BN statistics of a given sub-model are switch-independent."

As the last 0.25-width receives input from different paths of previous network, what does the word "independent/same statistics" mean?


2. Figure 4: what about Slimmable and Slimmable Wide IPKD for MobileNet experiments?

3. What is the training cost compared with vanilla training or US training, if all switches are enumerated in each iteration?

**Summary Of The Paper:**

The paper proposes ParaDiS, a slimmable network that can be executed across multiple devices, and be transferred across different devices without re-training. To train ParaDis, the authors propose to distill from a wider network, and combine activation knowledge distillation to improve the performance. While the idea of parallel execution is promising, there is little detailed discussion on that point.

**Summary Of The Review:**

Marginally below the acceptance threshold.

---

> ### Author Response · Authors · 2021-11-15
> **We have added new experiments on scaling up ParaDiS ResNet-50 with up to 8 thinner models and non trained switches coming "for free"**
>
> Thank you for your review.
>
> **Q4.1: Methodologically there is no fundamental differences from slimmable networks. The paper proposes 1) distilling from wider networks and 2) distilling feature maps, both of which are not new.**
>
> We agree that once the switches are defined there is no fundamental differences between ParaDiS and slimmable networks. However, we consider that defining the parallelizable ParaDiS switches as we did is original, inventive and has never been done before. Regarding points 1) and 2), we agree that they are not breakthrough inventions and they are based on existing concepts, but still, they include some originality. First, “distilling from wider networks” was considered before though not in the context of inplace knowledge distillation (IPKD) as we do in this paper. Second, “distilling feature maps” has been done as well, e.g., in [Bhardwaj2019] as we mention it, but again not in the context of IPKD.
>
> **Q4.2: There is little details on how to execute the parallel inference across multiple devices, which I think is a major issue of this paper. For instance: Does it mean there are three devices holding each part of the [0.5, 0.25, 0.25] configuration? If so, there is inevitable communication among devices.**
>
> Parallel inference across multiple devices is describes in the introduction: “Once a set of available devices is known, ...” Two options are possible: either (i) to load a suitable sub-model on each device, or (ii) to load the full model to each device and to execute the corresponding sub-model. Both options have their advantages. Sub-models are distributed as represented on Figure 1 (D). The only communication needed is to transmit the data to each device and then to retrieve at the end the result for a final fusion.
>
> **Q4.3: It would be more interesting to show the strategy to split the model across different devices, depending on their resources (memory or computation).**
>
> This is exactly what we are targeting: "Once a set of available devices is known, a suitable configuration may be selected ...". Indeed, here suitable may mean either in terms of memory or computation, or both. The choice between the last three options would depend on the particular use case.
>
> **Q4.4: No simulation results on parallel execution of ParaDis. I may still worry about the latency overhead comparing with the inference of a singe device, especially if different devices have different computation power, or there is any communication among devices.**
>
> Thank you for this remark, we fully agree that a graph showing the advantage of ParaDiS over US in terms of improved latency is necessary. We have added the necessary modifications to Figures 4 and 5 that are described in our answer to Q2.1 from another reviewer. As for devices with different computation power (or memory resources), switches with sub-models of different sizes like [0.5, 0.25, 0.25]× exist for this very purpose (see also answer to Q4.2).
>
> **Q4.5: It seems the width configurations are not scalable and sensitive to width configurations as discussed in Section 6. From to Figure 1, is it necessary that all neurons in the first width configuration are fully connected (in terms of the performance)?**
>
> For the first part of your question, see our answer to Q.3.3. We assume you are speaking about Figure 2. On this figure the fully connected layers are just represented for ease of understanding. In practice, the networks that we consider (e.g., MobileNet v1 and ResNet-50) consist mostly of different kinds of convolutional layers, except fully connected layers at the end for the classifiers.
>
> **Q4.6: It is still not clear to me the following statement: "Indeed, [...]" As the last 0.25-width receives input from different paths of previous network, what does the word "independent/same statistics" mean?**
>
> In this example the last 0.25-width does not receive anything from different paths of previous network. The only thing it receives is the input data. This is exactly what we mean, that the computations (the forward pass) that that happen in the last 0.25-width are totally independent from previous models. Though we have removed this discussion considering it not so important.
>
> **Q4.7: Figure 4: what about Slimmable and Slimmable Wide IPKD for MobileNet experiments?**
>
> We have not included Slimmable and Slimmable Wide IPKD results in Figure 4, since they are worse than the US Wide IPKD results.
>
> **Q4.8: What is the training cost compared with vanilla training or US training, if all switches are enumerated in each iteration?**
>
> Each sub-model update needs a forward/backward pass. Assuming forward/backward complexities of sub-models approximated as surfaces of squares as explained in Appendix A, the training complexities of Slimmable and ParaDiS examples given on the figure from Appendix A may be estimated as:
>
> - Vanilla training: 1.0
>
> - Slimmable training: 1^2 + 0.75^2 + 0.5^2 + 0.25^2 = 1.87
>
> - ParaDiS training: 1^2 + 2 x 0.5^2 + 4 x 0.25^2 = 1.75

---

### Official Review · Reviewer_fReP · 2021-11-02

**Correctness:** 3
**Technical Novelty And Significance:** 3
**Empirical Novelty And Significance:** 2
**Recommendation:** 6
**Confidence:** 2

**Main Review:**

Strengths
1. The experiment section is well written with clear description and thorough studies. The experimental results provide sound evidence to support claims made in the paper.

2. The proposed distillation algorithm is described clearly and straightforward to re-implement.

3. Described the corner cases like batch norm layers and proposed solutions to address them.

Weaknesses
1.  I hope this application of this work can be beyond the convolutional models in the image classification domain. Sharing all parameters across devices posed a serious constraints to those transformer-like models whose number of model parameters is much larger but not necessarily needs more compute flops per image.

2. The proposed framework required knowledge of all devices in advance before training. And it's not scalable when the list of switches is large.

3. The practice of distilling a bigger teacher to the student model is well established in the literature (like noisy student). It's better to add more related work along this line.

4. Another interesting baseline will be an ensemble of smaller models, each of which will be deployed on a separate device. In this way, the whole ensemble network can still work even after the number of devices in the network changes.



**Summary Of The Paper:**

The paper proposed an algorithm that distill from a larger teacher to a set of parallely distributable student models each of which only compute a fraction of channels in the network. The combined output of the student models can be used to compute the final class prediction.  In this way, student models can be deployed in a set of mobile or edge devices.

**Summary Of The Review:**

A careful study of distillation algorithm for the mobile/edge device deployment. The whole paper is well written with experimental results supporting the main contribution claimed by the authors. However, the significance of this approach is a bit limited due to the scalability issue (only works with convolutional models and  a very small set of switches.)

---

> ### Author Response · Authors · 2021-11-15
> **We have added new experiments on scaling up ParaDiS ResNet-50 with up to 8 thinner models and non trained switches coming "for free"**
>
> Thank you for your review. We did our best to consider your remarks and to answer your questions as below.
>
> **Q3.1: I hope this application of this work can be beyond the convolutional models in the image classification domain. Sharing all parameters across devices posed a serious constraints to those transformer-like models whose number of model parameters is much larger but not necessarily needs more compute flops per image.**
>
> Thank you for this remark. This approach might be very likely extended to Transformer-like models. In the current work we have only considered convolutional neural networks (CNNs). However, this approach is not at all restricted to some particular CNN architectures, but is applicable to most of modern CNN architectures.
>
> **Q3.2: The proposed framework required knowledge of all devices in advance before training.**
>
> We do not fully agree with this statement. Indeed, as we write in the introduction, the question we are trying to reply is “Can we have one neural network that may instantly and near-optimally be parallelly distributed over several devices, regardless of the number of devices and their capacities?” Thus, in fact we are trying to face a situation when we do not know the devices in advance: we do not know how many devices we will have, and we do not know their capacities. As such, every switch of a ParaDiS model targets an unknown configuration of devices. In contrast, the state-of-the-art approaches [Bhardwaj2019] and [Asif2019] we have mentioned in Section 2 “Related work” need to know the configuration of devices in advance before training.
>
> [Bhardwaj2019] Kartikeya Bhardwaj, Ching-Yi Lin, Anderson Sartor, and Radu Marculescu. Memory-and communication-aware model compression for distributed deep learning inference on iot. ACM Transactions on Embedded Computing Systems (TECS), 18(5s):1–22, 2019.
>
> [Asif2019] Umar Asif, Jianbin Tang, and Stefan Harrer. Ensemble knowledge distillation for learning improved and efficient networks. arXiv preprint arXiv:1909.08097, 2019.
>
> **Q3.3: And it's not [the framework is not] scalable when the list of switches is large.**
>
> Indeed, we have not yet managed to train 8 switches altogether as we show in Appendix F.2. However, we have obtained new results that are now added in Appendix F.1, where we train ResNet-50 while splitting it up to 8 thinner models (like [.125, ...], …). In summary, as described in Appendix F.1, in this setup we train only 5 switches to make then profit of 6 additional switches coming “for free”. This is another way to scale ParaDiS approach to more switches, and it seems viable. Indeed, the performance of this newly-trained model drops just slightly (less than 1 % of accuracy) as compared to the results from Figure 5 with much less switches.
>
> **Q3.4: The practice of distilling a bigger teacher to the student model is well established in the literature (like noisy student). It's better to add more related work along this line.**
>
> Thank you very much for this suggestion. We have added reference [Xie2020] to the introduction.
>
> [Xie2020] Xie, Q., Luong, M. T., Hovy, E., & Le, Q. V. (2020). Self-training with noisy student improves imagenet classification. In Proceedings of the IEEE/CVF Conference on Computer Vision and Pattern Recognition (pp. 10687-10698).
>
> **Q3.5: Another interesting baseline will be an ensemble of smaller models, each of which will be deployed on a separate device. In this way, the whole ensemble network can still work even after the number of devices in the network changes.**
>
> This is a very good suggestion, and we have already added a paper on ensembles [Wang2020] into Section 2 “Related work”. However, ensembles may only correspond to a subset of models having ParaDiS-like capacities. For example, a ParaDiS with switches [.25, .25, .25, .25], [.25, .25, .25], [.25, .25], and [.25] might be obtained via ensemble learning, but it cannot include bigger (e.g., .5) or smaller (e.g., .125) sub-models.
>
> [Wang2020] Xiaofang Wang, Dan Kondratyuk, Eric Christiansen, Kris M Kitani, Yair Movshovitz-Attias, and Elad Eban. On the surprising efficiency of committee-based models. arXiv preprint arXiv:2012.01988, 2020.

---

### Official Review · Reviewer_T9P1 · 2021-11-02

**Correctness:** 3
**Technical Novelty And Significance:** 2
**Empirical Novelty And Significance:** 1
**Recommendation:** 3
**Confidence:** 4

**Main Review:**

* Strength: Similar to slimmable networks, this approach could train all the networks at the same time and retraining is not necessary when deploying.
* Weakness: This approach fails to address several important problems in distributed embedded computing and the detailed advantages on hardware latency is missing.

* Reasons to accept:
  * The parallel distribution to speed up model deployment can help reduce latency.
  * The approach to train the ParaDiS network is computation-efficient and the performance is good.

* Reasons to reject:
  * Only a few experiments have been performed on classification task, without covering detection, segmentation etc. The ablation study could not show significant advantage of distillation.
   * Lack of specific information about how much latency is saved via parallel computing.

### Questions
* From Figure2, the second layer seems to depend on all the neurons of first layer, so how is it distributed in parallel? Is the computation done in a mixed serial-parallel style, or is the computation repeated?
* Many modern DLA have on-chip memory for storing intermediate results to save DDR bandwidth. Parallel computing across devices will introduce extra copies from and to DDR in this case. What's the cost model to decide when to distribute and when not to ?




**Summary Of The Paper:**

This paper proposed to train several parallel sub-networks in the slimmable networks framework, which could run in parallel on different devices subject to runtime hardware configurations to reduce latency.

**Summary Of The Review:**

The paper improves upon the slimmable network framework by introducing parallel sub-networks to reduce latency. However, the improvement has not been well empirically evaluated and justified. The main discussed scenario is distributed embedded computing, but important topics in this scenario like mapping between different traits of sub-network and the capabilities of devices, has not been covered. Not to mention runtime behaviors like load balancing and allocation of shared resources like CPU and memory. The paper may need be expanded significantly to include sufficient contents on the topic.

---

> ### Author Response · Authors · 2021-11-15
> **New task and new architecture: We have investigated ParaDiS for image super resolution task using the WDSR architecture.**
>
> Thank you for your review. We did our best to consider your remarks and to answer your questions as below.
>
> **Q2.1: This approach fails to address several important problems in distributed embedded computing and the detailed advantages on hardware latency is missing.**
>
> We agree that simulations on real hardware would be necessary to demonstrate the proposed approach. Our time and hardware constraints did not allow us to perform experiments in a realistic deployment, but we plan to perform measurements in realistic conditions with multiple devices as part of future work. Nevertheless, we also agree that the advantage of ParaDiS over US models in terms of latency improvement was not well demonstrated. We have improved on that in the updated version of the paper as follows. While in case of US it is quite easy to draw a graph of performance as a function switches since there is just one parameter that varies: the network width that is related to FLOPs. Doing the same is more difficult in case of ParaDiS since there is no unique strategy to describe all possible ParaDiS switches with just one parameter. Therefore, in the current version of the paper we have decided to represent the accuracy as a function of overall FLOPs. However, we totally agree that this representation does not demonstrate any gain of ParaDiS over US in terms of latency. Thus, we have added the following supplementary curves to the main graphs on Figures 4 and 5. We assume a situation when all the devices have equal capacities, though we have any number of them (of devices). In that case the US switch 0.5 and the ParaDiS switch [0.5, 0.5] will run with the same latency (neglecting the communication latency that is small) or the same number of FLOPs, while [0.5, 0.5] has a superior accuracy. The same holds for US switch 0.25 and ParaDiS switch [0.25, 0. 25, 0. 25, 0. 25]. This clearly represents the advantage of ParaDiS over US or Slimmable models. In summary, we have added on Figures 4 and 5, in addition to the blue curve, an orange curve that shows the accuracies of switches [0.25, 0. 25, 0. 25, 0. 25], [0.5, 0. 5], [1.0] as a function of FLOPs for one sub-model (and not sum of FLOPs for all sub-models as for the blue curve). Note that adding switch [0.5, 0. 25, 0. 25] to this curve does not make sense in this particular scenario since we assume that all the devices have equal capacities. On the same graph switch [0.5, 0. 25, 0. 25] will indicate worse accuracy than switch [0.5, 0.5] for the same number of biggest sub-model FLOPs. However, this does not mean either that the switch [0.5, 0. 25, 0. 25] is useless, it is useful in scenarios when the devices do not have equal capacities (contrary to what we assume in this particular evaluation).
>
> **Q2.2: Only a few experiments have been performed on classification task, without covering detection, segmentation etc.**
>
> Indeed, our experiments were only limited on ImageNet classification. Therefore, we have decided to investigate ParaDiS in application to image super resolution task and we have conducted experiments with Wide Activation Super-Resolution (WDSR) [Yu2020] ParaDiS model. Detailed result now added on Fig. 7 in Appendix D are inline with results on classification.
>
> [Yu2020] Yu, J., Y. Fan, and T. Huang. "Wide activation for efficient image and video super-resolution." 30th British Machine Vision Conference, BMVC 2019. 2020.
>
> **Q2.3: The ablation study could not show significant advantage of distillation.**
>
> We are not sure to well understand the ablation study the reviewer refers to. For example, one can clearly see from the figure from Appendix E.1 “Wide vs. non-wide and IPKD vs. no KD” that in case of MobileNet v1 the knowledge distillation (IPKD) improves the accuracy by 1 % in average, as compared to no distillation (no KD).
>
> **Q2.4: Lack of specific information about how much latency is saved via parallel computing.**
>
> We agree that the advantage of ParaDiS over US models in terms of latency improvement was not well demonstrated. We have addressed this issue in our answer to Q2.1.
>
> **Q2.5: From Figure2, the second layer seems to depend on all the neurons of first layer, so how is it distributed in parallel? Is the computation done in a mixed serial-parallel style, or is the computation repeated?**
>
> We apologize for the confusion. By gray circles on Figure 2 we represent either network inputs or its outputs. We have added the following sentence to the figure’s caption to clarify that: “While white circles denote the neurons, gray circles denote either network inputs (e.g., image pixels) or its outputs.”
>
> **Q2.6: Many modern DLA have on-chip memory for storing intermediate results to save DDR bandwidth. Parallel computing across devices will introduce extra copies from and to DDR in this case. What's the cost model to decide when to distribute and when not to ?**
>
> It is a very interesting suggestion. However, we have not addressed it yet, and we will study it in the future.

---

### Official Review · Reviewer_6dzy · 2021-11-05

**Correctness:** 3
**Technical Novelty And Significance:** 3
**Empirical Novelty And Significance:** 3
**Recommendation:** 5
**Confidence:** 4

**Main Review:**

=== Strengths ===

Figure 1 was very helpful in giving context to different types of distributed inference. Yet, on my first reading it wasn't clear whether the image was split spatially or channel-wise. This was clear though given my previous understanding of slimmable networks.

The ablation study was useful in confirming the importance of IPKD-A on top of US networks.

The inclusion of IPKD-A seems like a strong modification from the US networks.

The paper honestly discussed its limitations and difficulties scaling up to 8 switches.

=== Weaknesses ===

The major evaluation figures seem to not be highlighting the correct aspects of the method. In my understanding, this method should strive to maintain the accuracy of the single configurations as much as possible, while improving the model latency through parallelization. This is shown in the figures. Then, the comparison to US networks could either show improvements over previous work, or equality with improvements in latency or other aspects. This part seems to be lacking and currently from the evaluation itself it isn't clear why ParaDis would be preferred to US networks. US networks would also have decreased latency from their reduced width.

Ideally, the evaluation should include latency results since this is the biggest advantage of parallelization. This isn't directly captured by the FLOPS.

The ImageNet results are important, but there should be at least one table confirming that ParaDis works across other architectures. I imagine that the "information split" that ParaDis causes could have widely different effects in different networks. Residual branches, bottlenecks, and pointwise convolutions could possibly affect this.

Given the relatively poor single configuration results on some switches, these data points may want to be run multiple times and listed with error bars. It is hard to believe that cutting the network in half improves the ResNet accuracy up to two points.

I struggle to understand the statement: "within ParaDis networks, there might be potentially much more possible switches than in slimmable or US Nets." In US Networks, there is an ordering on the channels so that lowest index channels are the primary ones. This leads to up to C different configurations, since each switch only adds an extra channel. It seems like this should be true for ParaDis too, where the leftmost channels of the network are considered primary. If the network has 4 channels, then the possible configurations are [1], [.5, .5], [.5, .25, .25], and [.25, .25, .25, .25], and [.25, .25, .5] for example would not be valid. In that case, ParaDis also has at most C switches in general. In reality, just as in the US paper which limits the minimum channels to .25C, there would need to be a minimum sub-model size, leading to fewer than C switches.


=== Questions ===

Why is the [.5, .25, .25] configuration left untrained? I assume this was to demonstrate that it doesn't have to be explicitly trained to perform well, as long as its submodels are trained in other switches.

Other types of networks were mentioned in the related works, e.g. early exit models. How do these other types compare to ParaDis in terms of accuracy at different latencies?

If ParaDis and US Networks perform similarly for given compute, what situations would favor ParaDis models?

How are residual branches handled during the splits in ResNet?

What are the tradeoffs of updating all the switches vs. a subset of the switches per epoch? Did you choose to update all because in this case there were only 4 (or 3 if one of them wasn't trained)?

For the negative results for 8 switches, what is the minimum size of the channels in the split? I would be worried about the bottleneck of ResNet being split too much for example. Do these issues go away with a larger scaled width?

**Summary Of The Paper:**

The authors propose ParaDis, a parallel and distributed version of universally slimmable networks. They use the fundamentals from this work, plus a few additions, to train models that can be switched between different parallel factors with only minor accuracy degradation. They evaluate using ResNet and MobileNet on ImageNet against models trained solely for a given configuration to show that their technique matches and sometimes exceeds this baseline accuracy. They finish the paper with an ablation study that examines the impact of their additions to the training procedure.

**Summary Of The Review:**

This paper provides an interesting variant on the idea of slimmable networks that allows the technique to be applied to distributed inference. The idea is well-motivated and described in detail, yet the training procedure seems slightly ad-hoc and the evaluation falls short in multiple ways. The paper needs more baseline techniques, more model architectures, and larger parallelization factors for me to fully support acceptance.

---

> ### Author Response · Authors · 2021-11-15
> **We have added an analysis showing advantages of ParaDiS over US in terms of reduced latency**
>
> Thank you for your review.
>
> **Q1.1: Figure 1 was very helpful in giving context to different types of distributed inference. Yet, on my first reading it wasn't clear whether the image was split spatially or channel-wise. This was clear though given my previous understanding of slimmable networks.**
>
> We have understood that by “image” you mean the “feature maps”. Yes, indeed, they are split channel-wise as in slimmable networks. In addition, to make Figure 1 clearer, we have added to it an arrow indicating the network depth dimension.
>
> **Q1.2: The comparison to US networks could either show improvements over previous work, or equality with improvements in latency or other aspects. [...]**
>
> Thank you for this remark, we fully agree that a graph showing the advantage of ParaDiS over US in terms of improved latency is necessary. We have added necessary modifications to Figures 4 and 5 that are described in our answer to question Q2.1.
>
> **Q1.3: The ImageNet results are important, but there should be at least one table confirming that ParaDis works across other architectures. I imagine that the "information split" that ParaDis causes could have widely different effects in different networks. Residual branches, bottlenecks, and pointwise convolutions could possibly affect this.**
>
> Due to extensive computations needed, we have not applied ParaDiS to other architectures within ImageNet classification task. However, we have investigated ParaDiS in application to image super resolution task using the WDSR architecture (see our reply to Q2.2) that is quite different from MobileNet v1 and ResNet-50. Finally, we have investigated all various network elements you mentioned: residual branches are in ResNet-50 and WDSR, bottlenecks in ResNet-50, and pointwise convolutions in MobileNet v1.
>
> **Q1.4: Given the relatively poor single configuration results on some switches, these data points may want to be run multiple times and listed with error bars. It is hard to believe that cutting the network in half improves the ResNet accuracy up to two points.**
>
> We agree with you that those results should be stabilized. However, these additional experiments are very computationally demanding: computing the results for one single configurations requires up to 4 days on 4 GPUs. We had no time and resources to perform them since were running other additional experiments (see reply to Q2.2).
>
> **Q1.5: I struggle to understand the statement: "within ParaDis networks, there might be potentially much more possible switches than in slimmable or US Nets." [...].**
>
> We just meant that, e.g., with 4 channels for slimmable we have 1, .75, .5, .25; while for ParaDiS [1] , [.75, .25], [.75], [.5,.5], [.5,.25,.25], [.5,.25], [.25,.25,.25,.25], [.25,.25,.25] , [.25,.25] , [.25], and for 8 channels there are much more combinations for ParaDiS.
>
> **Q1.6: Why is the [.5, .25, .25] configuration left untrained? I assume this was to demonstrate that it doesn't have to be explicitly trained to perform well, as long as its submodels are trained in other switches.**
>
> Right, this is exactly why we left [.5, .25, .25] configuration untrained. We have also added additional experimental results to Fig. 11 in Appendix F.1, where we have trained ResNet-50 while splitting it up to 8 models: [.125, ...] (see reply to Q3.3).
>
> **Q1.7: Other types of networks were mentioned in the related works, e.g. early exit models. How do these other types compare to ParaDis in terms of accuracy at different latencies?**
>
> Since with ParaDiS we investigate the latency improvement thanks to model parallelization, we compare the results only with the baseline model that we parallelize, and not with all possible models.
>
> **Q1.8: If ParaDis and US Networks perform similarly for given compute, what situations would favor ParaDis models?**
>
> In case when several devices are available and it can be parallelized, see our reply to Q2.1.
>
> **Q1.9: How are residual branches handled during the splits in ResNet?**
>
> The residual branches are handled exactly as in slimmable networks.
>
> **Q1.10: What are the tradeoffs of updating all the switches vs. a subset of the switches per epoch? Did you choose to update all because in this case there were only 4 (or 3 if one of them wasn't trained)?**
>
> Yes, we have chosen to update them all, since there were only 3. Though in case of 8 switches (see Appendix F.2) we have also tried updating a random subset of switches per iteration, but this did not work.
>
> **Q1.11: For the negative results for 8 switches, what is the minimum size of the channels in the split? I would be worried about the bottleneck of ResNet being split too much for example. Do these issues go away with a larger scaled width?**
>
> We think this point was misunderstood. In Appendix F.2, we do not split the models into sub-models thinner than .25, we just start varying the span of the full switch, e.g., by including also switches like [.25, .25]. For sub-models thinner than .25, see our reply to Q3.3

---

### Author Response · Authors · 2021-11-15
**Thank you for your reviews.**

Dear reviewers, thank you again for the discussion. We have added to the paper some clarifications, additional experiments and additional analysis of results, as described in our detailed replies. We have uploaded a new updated version of the paper.

Changelog:

1. Adding an arrow to Figure 1 to clarify the network depth dimension (requested by R1).

2. Updating Figure 2 caption by indication that the gray cycles do not represent neurons but the inputs or outputs (requested by R2).

3. Adding an additional reference on knowledge distillation [Xie2020] Xie, Q., Luong, M. T., Hovy, E., & Le, Q. V. (2020). Self-training with noisy student improves imagenet classification. In Proceedings of the IEEE/CVF Conference on Computer Vision and Pattern Recognition (pp. 10687-10698) (requested by R3).

4. Additional results analysis:

- Adding analysis of ParaDiS over US advantage in terms of improved latency. Additional curves added to Figures 4 and 5, where we plot ParaDiS results as function of the maximum number of FLOPs instead of the sum of FLOPs. (requested by R1 and R2)

5. Additional experiments:

- Additional experiment on ParaDiS scaling. New results are added on Figure 11 in Appendix F.1, where we train ResNet-50 while splitting it up to 8 thinner models (like [.125, ...], …). In summary, as described in Appendix F.1, in this setup we train only 5 switches to make then profit of 6 additional switches coming “for free” (requested by R1, R3 and R4

- New task and new architecture: Investigating ParaDiS for image super resolution task using the WDSR architecture. These results are added on Figure 7 in a new Appendix D (requested by R1 and R2).

---

### Decision · Program_Chairs · 2022-01-20

**Decision:**

Reject

**Comment:**

This paper presents an approach for distilling a larger teacher model into a set of students that can run in parallel at lower cost. The main strengths are that the approach appears conceptually sound and reasonably well executed. The main weaknesses are that the differences relative to previous work is fairly slim, and the experimental results are overly idealized. While the main benefit of the approach is improvement in latency, the experiments evaluate in terms of FLOPS. There was some back and forth between the authors and reviewers about these points. Authors added some additional results and seem to acknowledge the limitations, e.g., saying “Our time and hardware constraints did not allow us to perform experiments in a realistic deployment.” However, for a paper primarily concerned with reducing latency, reviewers were unconvinced that this evaluation was sufficient.